# School bullying and its association with psychological wellbeing: Findings of the Fujian Adolescent Mental Wellness Study (FAMWeS)

Jian Jiang[1,2*], Zhijie Luo[3], Yuhang Chen[3], Haridah Alias[4], Li Ping Wong[3,4*], Yulan Lin[4*], Zhijian Hu[3*]

1 Fuzhou University Affiliated Provincial Hospital, Shengli Clinical Medical College of Fujian Medical University, Fuzhou, China, 2 School of Health Management, Fujian Medical University, Fuzhou, China, 3 Fujian Key Laboratory of Environmental Factors and Cancer, Department of Epidemiology and Health Statistics, School of Public Health, Fujian Medical University, Fuzhou, China, 4 Centre of Population Heath (CePH), Department of Social and Preventive Medicine, Faculty of Medicine, Universiti Malaya, Kuala Lumpur, Malaysia

* 52667428@qq.com, wonglp@ummc.edu.my, yulanlin@fjmu.edu.cn, huzhijian@fjmu.edu.cn

## Abstract

### Background

School bullying is becoming a prevalent phenomenon in China, posing a growing threat to the psychological well-being of children and adolescents. This study aims to examine bullying victimization and bullying behavior using data from FAMWeS (the Fujian Adolescent Mental Wellness Study), and how they impact emotional state and psychological distress.

### Method

The study conducted in entire Fujian Province, China, involved a cross-sectional questionnaire survey on adolescent students from secondary and high schools, employing multi-stage stratified cluster sampling methods from May to July 2023. Primary outcome measures were bullying victimization and bullying behavior. Emotional state was evaluated using the Depression, Anxiety, and Stress Scale for Youth (DASS-Y), and psychological distress was assessed using the General Health Questionnaire-12 (GHQ-12).

### Results

Analysis of 53,157 valid responses revealed that 19.4% of respondents reported being victims of bullying, 9.9% reported engaging in bullying behavior, and 8.2% reported experiencing both bullying victimization and engaging in bullying themselves. A minority were found to have depressive (19.1%), anxiety (22.0%), and stress (9.2%) symptoms. The presence of psychological distress was observed in 54.4% of the participants (GHQ score of 3−12). Significant associations were found

**Data availability statement:** Due to the privacy concerns of adolescents, and for ethical and confidentiality reasons, the research data cannot be shared. For interested researchers the data may be obtained from the corresponding authors or the Ethics Committee of Fuzhou University Affiliated Provincial Hospital (E-mail: fjslkyk@163.com).

**Funding:** This study was supported by the Scientific and Technological Innovation Joint Capital Projects of Fujian Province (No.2020Y9018 to Zhijian Hu), the Natural Science Foundation of Fujian Province (No. 2021J01726, No. 2021J01733 to Zhijian Hu), the Central government-led local science and technology development special project (No. 2020L3009, No. 2019L3006 to Zhijian Hu), and the Fujian Provincial Health and Wellness Subsidy Project（2025876 to Jian Jiang).

**Competing interests:** The authors have declared that no competing interests exist.

between bullying victimization and the DASS-Y subscales of depression, anxiety, and stress. Odds ratios indicated a 1.73 times higher likelihood for depression (OR 1.73, 95% CI 1.521.96), 2.24 times higher for anxiety (OR 2.24, 95% CI 1.96–2.55), and 1.59 times higher for stress (OR 1.59, 95% CI 1.38–1.84). Similarly, psychological distress measured by the GHQ was significantly associated with bullying victimization (OR 1.77, 95% CI 1.68–1.86). Bullying behavior also showed significant associations with all DASS-Y subscales and psychological distress. Geographic disparities revealed varying rates across districts, while factors such as age, academic performance, and parental education emerged as significant influences on bullying victimization and behavior.

## Conclusion

The findings underscore a concerning prevalence of bullying victimization and behavior, highlighting the critical need for comprehensive anti-bullying strategies that address both victimization and perpetration dynamics. There is also a clear need for interventions targeting socio-economic disparities across various geographic regions.

---

## 1. Introduction

School bullying has been recognized as a substantial public health and social concern among adolescents of school age worldwide [1]. According to UNICEF, globally, slightly over one-third of students aged 13–15 experience bullying, and three in ten students across 39 industrialized countries admit to bullying their peers [2]. Likewise in China, school bullying represents a significant societal concern. One finding revealed that 26.1% of students from all levels of pre-college education—primary, middle, high and vocational school—reported instances of bullying victimization [3]. A recent nationwide study reported a bullying victimization rate of 10.9% [4] while another study found that 25.7% of middle school students reported being bullied [5]. A recent study from Henan Province highlighted an even higher rate, with 42.4% of Chinese high school students reported experiencing bullying [6].

The ramifications of school bullying extend beyond individual suffering. Among the most widely reported consequences are mental and psychological impacts. Bullying has been widely reported to result in depressive symptoms and various other psychological challenges. Victims of bullying are more likely to develop long-lasting emotional adjustment issues, such as depression, anxiety, withdrawal, and loneliness [7] which potentially persist for the rest of their lives. The profound psychological consequences of bullying are also highlighted by a recent meta-analysis, which found that children and adolescents who experienced bullying were nearly three times more likely to develop depression compared to those who did not face bullying [8]. Similarly evident in China, a study from Henan Province revealed alarming findings, where a substantial 42.4% of school students experienced bullying, the prevalence of depression was 30.0% [6].

Beyond mental and psychological impact, childhood bullying also exacerbates the probability of adverse social, and educational outcomes [9,10]. Research conducted in China revealed that school bullying diminishes peer cooperation by eroding students' feelings of belonging within the school environment [11]. In a study spanning 51 countries, evidence pointed to the adverse effects of bullying victimization on academic literacy and social integration [12]. However, bullying victims often feel unsafe at school and isolated from these interactions, resulting in decreased engagement in school activities and negatively affecting classroom participation and academic performance [10,13]. If unaddressed, bullying incidents could escalate, heightening tensions and jeopardizing the well-being of both individual students and the school community. Therefore, addressing school bullying is not just protecting individual mental health but also about preserving social cohesion and stability.

Research has primarily focused on the consequences for those who are bullied, with limited investigation into the perpetrators of school bullying. Accurately assessing bullying behavior poses a challenge due to the difficulty in obtaining admissions of such behavior from students. In China, a study among vocational school students (equivalent to senior secondary school) found that 30.4% reported being bullied, 2.9% reported bullying others, and 21.7% reported both being bullied and bullying others [14]. Studies indicate that individuals who both bully and are bullied face a unique set of consequences, including a combination of externalizing and internalizing problems, negative self-perception, and peer rejection [9].

The Fujian Adolescent Mental Wellness Study (FAMWeS) is a comprehensive survey of schoolchildren in Fujian province, focusing on various aspects of adolescent mental health, lifestyles, social issues, and general wellbeing. Among the variables assessed were victimization and bullying behaviors. This study primary aims to explore the demographics, family environments, and psychological well-being associated with both bullying victimization and bullying behavior. Additionally, it investigates participants' feelings of safety at school and their association with both bullying victimization and bullying behavior. The findings from this large-scale provincial study will provide valuable insights to gauge the prevalence of school bullying and its psychosocial impact on children in Fujian province, China. The comprehensive data collected across this province will help identify the scope of the bullying problem and guide the implementation of evidence-based strategies to address this significant public health and social issue affecting school-aged youth.

## 2. Methodology

### 2.1. Sample and setting

The study was conducted in Fujian Province, China, which is situated on the southeast coast of the country. Fujian consists of nine cities: Fuzhou, Xiamen, Zhangzhou, Quanzhou, Sanming, Putian, Nanping, Longyan, and Ningde, along with 9 county-level cities, 31 municipal districts, and 42 counties. By the end of 2022, Fujian had 1,832 registered secondary schools, including 1,273 junior high schools (grades 7–9) and 559 senior high schools (grades 10–12). The total number of registered students was 22,253,397, with 1,526,120 in junior high schools and 699,277 in senior high schools. From May to July 2023, a cross-sectional questionnaire survey on mental health was conducted on adolescent students enrolled in secondary and high schools in Fujian Province using multi-stage stratified cluster sampling method.

The sampling for the study involved three stages. In the first stage, Fujian Province was divided into three economic development levels based on the 2022 economic development data for all prefecture-level citied published by the Fujian Provincial Bureau of Statistics, and two cities were randomly selected from each level, resulting in a total of six cities: Fuzhou, Zhangzhou, Ningde, Longyan, Putian, and Nanping. Economically high-developed regions included Fuzhou City (Eastern Fujian), Xiamen City, Quanzhou City, and Zhangzhou City (Southern Fujian). Medium-developed regions included Ningde City (Eastern Fujian) and Longyan City (Western Fujian). Underdeveloped regions included Putian City (Central Fujian), Nanping City, and Sanming City (Northern Fujian).

A total of six prefecture-level cities were selected for the survey: Fuzhou City (1 county-level city, 6 municipal districts, and 6 counties), Zhangzhou City (4 municipal districts and 7 counties), Ningde City (2 county-level cities, 1 municipal

district, and 6 counties), Longyan City (1 county-level city, 3 municipal districts, and 4 counties), Putian City (4 municipal districts and 1 county), and Nanping City (3 county-level cities, 2 municipal districts, and 5 counties). All the administrative counties (cities, districts) under these selected prefecture-level cities were included in the survey. In the second stage, a list of secondary and high schools in the selected cities was obtained, and schools were categorized accordingly. Using simple random sampling, one secondary school and one high school were selected from each administrative area. In the third stage, cluster sampling was used to select at least 30% of classes from each grade (1st, 2nd, and 3rd) within the chosen schools, with all students in the selected classes participating in the study.

## 2.2. Sample size calculation

The sample size is calculated using the formula $n = $ design effect (DEFF) $u^2$ a/2 p(1-p)/$d^2$ [15] with a confidence level of 95% and a corresponding u-value of 1.96. The design effect (DEFF) is set at 1.5, and d is defined as 10% of p. Based on previous studies, the probability (p) of mental health problems is assumed to be 20%. The estimated sample size for each stratum is 7,540. Given there are 6 stratum (6 grades), the minimum required sample size is 45,240.

## 2.3. Data collection

The study involved a systematic process wherein teachers in selected classes facilitated the completion of paper questionnaires by students during class time. Initially, the research team liaised with education administration departments across six cities, outlining the study's objectives, significance, and expected outcomes to secure their support. These departments facilitated initial communication and arrangements with the chosen schools. Subsequently, the research team directly engaged with principals and teachers, providing insights into the study's background, requirements, and procedures, along with the necessary school cooperation. Teachers underwent two online training sessions to clarify questionnaire content and classroom protocols. Electronic copies of the questionnaires were sent to schools for printing and distribution, with classroom teachers overseeing the distribution, completion, and collection processes. All students in the selected classes were provided with consent forms to fill out, and every student returned the form, indicating their willingness to participate. Without any refusals or withdrawals, all selected students participated in the questionnaire, which typically took around 20 minutes to complete.

To ensure valid and reliable responses, the research team implements quality control measures, including two trap questions within the questionnaire to identify inconsistent responses. Firstly, responses to Question 1.15 ("Do you have siblings?") and Question 1.31 ("How many siblings do you have, excluding yourself?") are scrutinized. If a participant answers "No" to Question 1.15 but indicates having one or more siblings in Question 1.31, the questionnaire is deemed invalid and excluded from analysis. Secondly, responses to Question 1.2 ("Where do you come from, current place of residence?") and Question 1.45 ("What is your current place of residence?") are assessed. Although both questions solicit information regarding rural, urban, and county residences, variations in the order of these responses indicate inconsistency. Consequently, surveys providing discrepant responses to these questions are also eliminated from consideration, ensuring the integrity of the dataset. Additionally, any responses exhibiting straight-lining, where all answers were selected in the same manner, are also removed.

## 2.4. Instrument

The FAMWeS is an extensive examination of schoolchildren in Fujian province, delving into diverse facets of adolescent mental health, lifestyles, family dynamics, and overall wellbeing. This report utilizes data from the FAMWeS, considering variables such as demographics, emotional state, and psychological distress as independent variables. Demographic data includes age, gender, location, academic performance, maternal and paternal educational levels, and family composition. The analysis focuses on outcomes namely bullying victimization, bullying behavior, and feelings of safety at school as key measures.

The emotional state was evaluate using Depression, Anxiety, and Stress Scale for Youth (DASS-Y). The DASS-Y is a version of the Depression, Anxiety, and Stress Scale – 21 items (DASS-21) for youth aged 7–18 years of age designed to measure the negative emotional states of depression, anxiety and stress [16]. The cutoff scores for the DASS-Y were determined using the same percentile ranges as those for the adult DASS. These cutoffs classify scores into levels of normal, mild or moderate, and severe or extremely severe for depression, anxiety, and stress [16]. The Chinese version of the DASS-Y was utilized, and it showed superior robustness compared to the DASS-21 in terms of psychometric properties [17].

Psychological distress was assessed using the validated Chinese version of GHQ-12 [18]. The validated Chinese version of the GHQ-12 has shown robust reliability and validity in the Chinese population and has been extensively utilized and is widely recognized within the population [18–20]. The GHQ-12 measure includes standardized instructions and scoring interpretations. It is administered as a self-report, where the participants answered 12 questions about their personal life over the past few weeks. Method of scoring is using the bimodal (0-0-1-1) approach, where responses to all items are summed to yield a total score ranging from 0 to 12, with higher scores indicating greater psychological distress. The total GHQ-12 score ranges from 0 to 12. The scores were categorized into 0–2 (no distress) and 3–12 (psychological distress) [21].

The bullying victimization question assessed whether participants had ever experienced being bullied or teased by classmates at school. The bullying behavior question inquired whether participants had ever engaged in bullying or teasing others at school. Response options for both questions included "yes" and "no." The question regarding feeling safe at school asked participants to indicate their perception of safety, with response options being "unsafe," "neutral," and "safe."

## 2.5. Ethical consideration

This study was approved by the Ethics Committee of Fujian Provincial Hospital (Lun Audit Research No. K2023-03–005) and has been registered with the China Clinical Trial Registry (https://www.chictr.org.cn) (registration number: ChiCTR2300075694). Informed consent was obtained from parents or legal guardians. Consent forms, which mandated signatures from both students and their parents or guardians, were disseminated to selected students in the sampling plan by classroom teachers. The consent form clearly indicated voluntary participation and assured confidentiality of responses. Only those who returned a completed consent form were provided with the questionnaire for completion. The study ensured the privacy and confidentiality of participants, with all data anonymized to prevent identification. Personal information was kept secure, and responses were coded to maintain anonymity, safeguarding participants' identities and ensuring ethical compliance.

## 2.6. Inclusivity in global research

Additional information regarding the ethical, cultural, and scientific considerations specific to inclusivity in global research is included in the Supporting Information (S1 Checklist)

## 2.7. Statistical analyses

Cronbach's alpha coefficient was also determined to evaluate the internal consistency of DASS-Y. In this study, Cronbach's $\alpha$ for three subscales of DASS-Y were 0.894 for depression, 0.864 for anxiety, and 0.870 for stress. Cronbach's $\alpha$ for GHQ-12 was 0.626. All scales demonstrated good reliability, with an alpha coefficient between 0.6 and 0.7 indicating an acceptable level of reliability, and 0.8 or greater indicating a very good level [22]. Descriptive statistics were calculated for the sample demographic characteristics and the independent variables. The categorical variables are presented as frequencies and percentages, whereas the quantitative variables are presented as means and standard deviations when they are normally distributed; otherwise, they are shown as medians and interquartile ranges (IQRs). Univariate and multivariable analyses were employed to determine the factors (demographics and emotional states) associated with bullying

victimization and bullying behavior. All factors that gave results in the univariate analyses below the significance level of 0.05 were included in a multivariate analysis. Hosmer–Lemeshow goodness-of-fit tests were used to determine the fitness of the model. Statistical significance was established at a *p*-value < 0.05. All analyses were also conducted using SPSS version 22.0 (SPSS Inc., Chicago, IL, USA).

## 3. Results

A total of 54,368 questionnaires were collected. After data cleaning, 53,157 valid responses were included in the final analysis. As shown in Table 1, the age distribution of participants showed that the majority were aged 11–14 years (40.9%) and 15–16 years (38.0%). There was an almost equal proportion of males (51.1%) and females (48.9%). Regarding locality, the majority were from rural areas (45.6%) and towns (39.3%). The majority of participants reported that the highest paternal education level was high school or college (37.2%), and similarly, the highest maternal education level was high school or college (35.9%). In terms of family composition, the majority were from intact families (90.3%).

The descriptive results of the DASS-Y for all participants are presented in the second column of Table 1. Overall, the majority of respondents exhibited normal levels of depression (80.9%), anxiety (78.0%), and stress (90.8%). However, a notable minority experienced mild to moderate or severe to extremely severe levels of depressive symptoms (19.1%), anxiety (22.0%), and stress (9.2%). The findings from the GHQ-12 indicate that slightly over half of the participants (54.4%) reported a GHQ score of 3–12, suggesting the presence of psychological distress.

### 3.1. Bullying victimization

Fig 1 shows that 19.4% of respondents reported experiencing bullying victimization. By district, Zhangzhou had the highest proportion of respondents reporting bullying victimization (29.6%), followed by Longyan (22.5%) and Quanzhou (22.5%). The lowest proportions were in Fuzhou (13.3%) and Putian (13.4%).

As shown in Table 1, multivariable analysis revealed a significant gradual increase in bullying victimization from older to younger age groups. Among participants aged 12−14 years, 21.0% reported bullying victimization. The odds of bullying victimization were 1.68 times greater for individuals aged 12−14 compared to those aged 17−19 (OR 1.68, 95% CI 1.58–1.79). Males reported a significantly higher proportion of bullying victimization (OR 1.66, 95% CI 1.59–1.74). Individuals from rural areas exhibited a greater likelihood of bullying victimization compared to those from city areas (OR 1.35, 95% CI 1.26–1.46). There was a significant and gradual increase in bullying victimization as academic performance declined. The odds of bullying victimization were 1.68 times higher for participants with D-level performance (OR 1.68, 95% CI 1.29–1.51) and 1.40 times higher for those with E-level performance (OR 1.40, 95% CI 1.28–1.53) compared to those in the top 15% (A level).

Participants whose parents had postgraduate degrees reported the lowest proportion of bullying victimization, with rates of 17.4% for paternal education and 17.6% for maternal education. However, in the multivariable model, only maternal education level was a significant predictor of bullying victimization. Individuals whose fathers had undergraduate degree had higher odds of bullying victimization compared to those whose fathers had postgraduate degrees (OR 1.17, 95% CI 1.02–1.35).

Multivariable analysis also revealed a significant association between the depression, anxiety, and stress subscales of the DASS-Y and bullying victimization. The odds of bullying victimization gradually increased with the severity of the subscale. For the depression subscale, individuals with severe or extremely severe depression had 1.73 times higher odds of bullying victimization compared to those with normal depression (OR 1.73, 95% CI 1.521.96). For the anxiety subscale, individuals with severe or extremely severe anxiety had 2.24 times higher odds of bullying victimization compared to those with normal anxiety (OR 2.24, 95% CI 1.96−2.56). For the stress subscale, individuals with severe or extremely severe stress had 1.59 times higher odds of bullying victimization compared to those with normal stress (OR 1.59, 95% CI 1.38–1.84). Additionally, multivariable analysis indicated a significant association between psychological distress and

**Table 1. Factors associated with bullying victimization and bullying behavior (N = 53157).**

| | | Bullying victimization | | | Bullying behavior | | |
| --- | --- | --- | --- | --- | --- | --- | --- |
| | | Univariable analysis | | Multivariable analysis | Univariable analysis | | Multivariable analysis |
| | N (%) | Ever being bullied or teased by classmates at school (n = 10324) | p-value | Ever being bullied classmates at school vs Never being bullied classmates at school | Ever bullied or teased other students at school (n = 5272) | p-value | Ever bullied or teased other students at school vs Never bullied or teased other students at school |
| | | | | OR (95% CI) | | | OR (95% CI) |
| Age (years) | | | | | | | |
| 12-14 | 21733 (40.9) | 4574 (21.0) | p<0.001 | 1.68 (1.58-1.79)*** | 2073 (9.5) | P=0.001 | 1.10 (1.02-1.19)* |
| 15-16 | 20220 (38.0) | 4044 (20.0) | | 1.45 (1.36-1.55)*** | 2134 (10.6) | | 1.15 (1.06-1.25)*** |
| 17-19 | 11204 (21.1) | 1706 (15.2) | | Reference | 1065 (9.5) | | Reference |
| Gender | | | | | | | |
| Male | 27137 (51.1) | 6012 (22.2) | p<0.001 | 1.66 (1.59-1.74)*** | 3490 (12.9) | p<0.001 | 2.18 (2.05-2.31)*** |
| Female | 26020 (48.9) | 4312 (16.6) | | Reference | 1782 (6.8) | | Reference |
| Location | | | | | | | |
| Rural | 24223 (45.6) | 5460 (22.5) | p<0.001 | 1.35 (1.26-1.46)*** | 2993 (12.4) | p<0.001 | 1.41 (1.29-1.55)*** |
| Town | 20879 (39.3) | 3405 (16.3) | | 0.92 (0.86-0.99)* | 1514 (7.3) | | 0.80 (0.72-0.88)*** |
| City | 8055 (15.2) | 1459 (18.1) | | Reference | 765 (9.5) | | Reference |
| Academic performance | | | | | | | |
| Top 15% (A level) | 8932 (16.8) | 1425 (16.0) | p<0.001 | Reference | 787 (8.8) | p<0.001 | Reference |
| 15%−30% (B level) | 12664 (23.8) | 2177 (17.2) | | 1.08 (1.00-1.17)* | 1034 (8.2) | | 0.92 (0.83-1.102) |
| 30%−50% (C level) | 15779 (29.7) | 2993 (19.0) | | 1.19 (1.10-1.28)*** | 1485 (9.4) | | 1.07 (0.97-1.17) |
| 50%−75% (D level) | 10938 (20.6) | 2478 (22.7) | | 1.68 (1.29-1.51)*** | 1233 (11.3) | | 1.22 (1.10-1.34)*** |
| >75% (E level) | 4844 (9.1) | 1251 (25.8) | | 1.40 (1.28-1.53)*** | 733 (15.1) | | 1.40 (1.25-1.57)*** |
| Maternal highest education level | | | | | | | |
| Elementary School | 7466 (14.0) | 1596 (21.4) | p<0.001 | 1.09 (0.94-1.26) | 916 (12.3) | p<0.001 | 1.12 (0.92-1.35) |
| Junior High School | 10500 (19.8) | 2042 (19.4) | | 1.03 (0.90-1.18) | 956 (9.1) | | 0.89 (0.75-1.06) |
| High School or Junior College | 19109 (35.9) | 3657 (19.1) | | 1.06 (0.94-1.20) | 1772 (9.3) | | 0.97 (0.83-1.15) |
| Post-Secondary | 9251 (17.4) | 1771 (19.1) | | 1.09 (0.96-1.23) | 974 (10.5) | | 1.11 (0.94-1.30) |
| Undergraduate | 3285 (6.2) | 640 (19.5) | | 1.17 (1.02-1.35)* | 348 (10.6) | | 1.21 (1.01-1.44)* |
| Postgraduates | 3546 (6.7) | 618 (17.4) | | Reference | 306 (8.6) | | Reference |
| Paternal highest education level | | | | | | | |
| Elementary School | 7389 (13.9) | 1552 (21.0) | p<0.001 | 0.94 (0.82-1.08) | 882 (11.9) | p<0.001 | 1.08 (0.91-1.29) |
| Junior High School | 7743 (14.6) | 1501 (19.4) | | 0.89 (0.78-1.01) | 721 (9.3) | | 0.92 (0.78-1.09) |
| High School or Junior College | 19778 (37.2) | 3797 (19.2) | | 0.91 (0.82-1.03) | 1787 (9.0) | | 0.91 (0.78-1.06) |
| Post-Secondary | 10178 (19.1) | 1984 (19.5) | | 1.02 (0.91-1.14) | 1116 (11.0) | | 1.19 (1.02-1.38)* |
| Undergraduate | 3515 (6.6) | 690 (19.6) | | 1.07 (0.91-1.14) | 375 (10.7) | | 1.20 (1.01-1.41)* |
| Postgraduates | 4554 (8.6) | 800 (17.6) | | Reference | 391 (8.6) | | Reference |
| Family composition | | | | | | | |
| Intact Families | 47986 (90.3) | 9127 (19.0) | p<0.001 | Reference | 4709 (9.8) | P=0.033 | Reference |
| Single-parent families | 1763 (3.3) | 412 (23.4) | | 1.11 (0.98-1.25) | 197 (11.2) | | 1.00 (0.86-1.17) |
| Divorced families | 2556 (4.8) | 574 (22.5) | | 1.01 (0.92-1.12) | 259 (10.1) | | 0.92 (0.80-1.05) |
| Widowed families | 441 (0.8) | 98 (22.2) | | 1.10 (0.87-1.40) | 56 (12.7) | | 1.26 (0.94-1.68) |
| Intergenerational families | 441 (0.8) | 113 (27.5) | | 1.24 (0.98-1.56) | 51 (12.4) | | 1.01 (0.74-1.37) |

*(Continued)*

**Table 1.** (Continued)

| | | Bullying victimization | | | | Bullying behavior | | |
|---|---|---|---|---|---|---|---|---|
| Emotional state/DASS-Y | | | | | | | | |
| Depression | | | | | | | | |
| Normal | 43014 (80.9) | 6793 (15.8) | p<0.001 | Reference | | 3608 (8.4) | p<0.001 | Reference |
| Mild/ Moderate | 8232 (15.5) | 2618 (31.8) | | 1.43 (1.33-1.52)*** | | 1294 (15.7) | | 1.32 (1.21-1.43)*** |
| Severe/ Extremely severe | 1911 (3.6) | 913 (47.8) | | 1.73 (1.52-1.96)*** | | 370 (19.4) | | 1.25 (1.06-1.47)** |
| Anxiety | | | | | | | | |
| Normal | 41486 (78.0) | 6367 (15.3) | p<0.001 | Reference | | 3419 (8.2) | p<0.001 | Reference |
| Mild/ Moderate | 9868 (18.6) | 3086 (31.3) | | 1.73 (1.62-1.84)*** | | 1498 (15.2) | | 1.53 (1.41-1.66)*** |
| Severe/ Extremely severe | 1803 (3.4) | 871 (48.3) | | 2.24 (1.96-2.56)*** | | 355 (19.7) | | 1.66 (1.40-1.97)*** |
| Stress | | | | | | | | |
| Normal | 48266 (90.8) | 8401 (17.4) | p<0.001 | Reference | | 4453 (9.2) | p<0.001 | Reference |
| Mild/ Moderate | 3604 (6.8) | 1299 (36.0) | | 1.38 (1.27-1.50)*** | | 557 (15.5) | | 1.16 (1.04-1.30)** |
| Severe/ Extremely severe | 1287 (2.4) | 624 (48.5) | | 1.59 (1.38-1.84)*** | | 262 (20.4) | | 1.34 (1.12-1.61)** |
| Psychological distress/ GHQ-12 | | | | | | | | |
| No psychological distress (0–2) | 24250 (45.6) | 3031 (12.5) | p<0.001 | Reference | | 1585 (6.5) | p<0.001 | Reference |
| Psychological distress (3–12) | 28907 (54.4) | 7293 (25.2) | | 1.77 (1.68-1.86)*** | | 3687 (12.8) | | 1.71 (1.60-1.83)*** |

\* *p*<0.05, \*\* *p*<0.01, \*\*\* *p*<0.001;

[a] Hosmer–Lemeshow test, chi-square: 14.839, *p*-value: 0.062; Nagelkerke R²: 0.118

[b] Hosmer–Lemeshow test, chi-square: 20.045, *p*-value: 0.010; Nagelkerke R²: 0.083

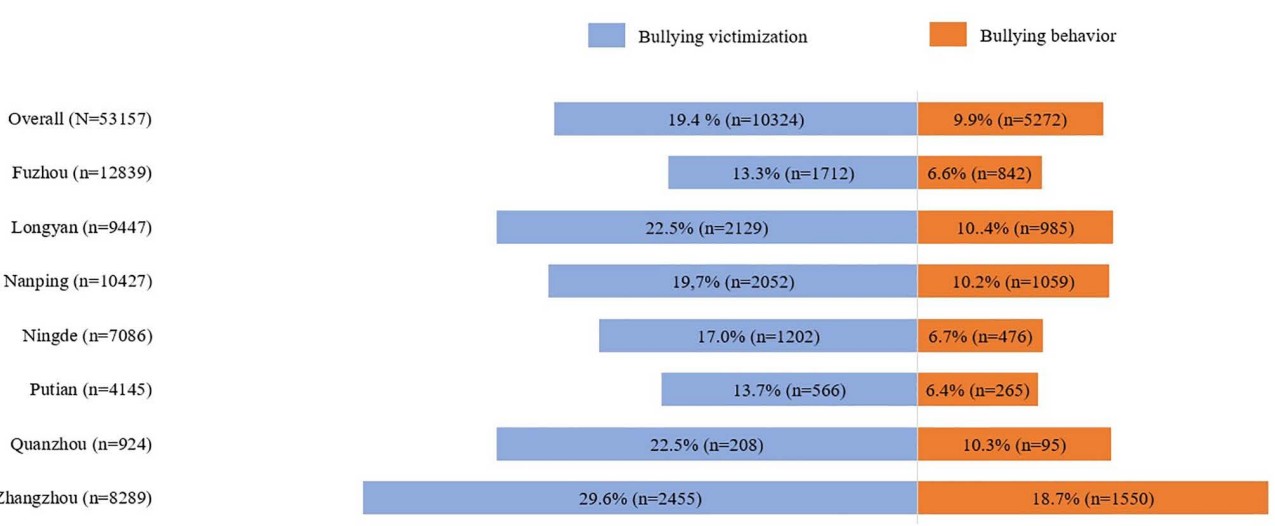

**Fig 1. Proportion of bullying victimization and bullying behavior (N = 53157).**

bullying victimization. Individuals with a GHQ score of 3−12 had 1.77 times higher odds of bullying victimization compared to those with a score of 0−2 (OR 1.77, 95% CI 1.68–1.86).

### 3.2. Bullying behavior

Overall, 9.9% of respondents reported engaging in bullying behavior, and 8.2% reported experiencing both being bullied and engaging in bullying themselves. By district, Zhangzhou had the highest proportion of respondents reporting bullying behavior (18.7%), followed by Longyan (10.4%) and Quanzhou (10.3%). The lowest proportions were in Putian (6.4%) and Fuzhou (6.6%).

As shown in Table 1, multivariable analysis revealed that the odds of bullying behavior were higher in participants aged 12−14 (OR 1.10, 95% CI 1.02–1.19) and 15–16 (OR 1.15, 95% CI 1.06–1.25) compared to those aged 17–19. Males reported a significantly higher proportion of bullying behavior (OR 2.18, 95% CI 2.05–2.31). There was no specific trend in bullying behavior by location. The highest rates of bullying behavior were reported by individuals from rural areas. Individuals from rural locations exhibited a significantly greater likelihood of bullying behavior compared to those from city areas (OR 1.41, 95% CI 1.29–1.55). However, individuals from town areas reported a significantly lower likelihood of bullying behavior compared to those from city areas (OR 0.80, 95% CI 0.72–0.88. Based on academic achievement, the odds of engaging in bullying behavior were 1.22 times higher for participants with D-level performance (OR 1.22, 95% CI 1.10–1.34) and 1.40 times higher for those with E-level performance (OR 1.40, 95% CI 1.25–1.57) compared to those in the top 15% (A level).

Participants whose parents had postgraduate degrees reported the lowest incidence of bullying behavior. In the multivariable model, individuals whose mothers had an undergraduate degree had a significantly higher likelihood of exhibiting bullying behavior compared to those whose mothers had a postgraduate degree (OR 1.21, 95% CI 1.01–1.44). Similarly, individuals whose fathers had post-secondary (OR 1.19, 95% CI 1.02–1.38) or undergraduate (OR 1.20, 95% CI 1.01–1.41) degrees reported a significantly higher likelihood of bullying behavior compared to those whose fathers had postgraduate degrees.

A significant association was found between the depression, anxiety, and stress subscales of the DASS-Y and bullying behavior. In all three subscales, individuals with severe or extremely severe, as well as mild or moderate levels, reported a significantly higher likelihood of bullying behavior compared to those with normal levels. A significant association between psychological distress and bullying behavior was also identified. Individuals with a GHQ score of 3−12 had 1.71 times higher odds of bullying behavior compared to those with a score of 0−2 (OR 1.71, 95% CI 1.60–1.83).

### 3.3. Feeling safe at school

Fig 2 illustrates the proportion of individuals feeling safe at school across different districts. Overall, 57.4% of respondents reported feeling safe, 38.6% reported being neutral, and only 4.0% reported feeling unsafe. The highest proportion of respondents feeling unsafe was from Zhangzhou (5.2%), followed by Quanzhou (4.8%), with the lowest proportions from Fuzhou (3.1%) and Putian (3.8%). The associated factors with feeling safe in school were shown in Table 2. Participants who experience bullying victimization have higher odds of feeling unsafe at school (OR 4.60, 95% CI 4.14–5.11). Similarly, participants with bullying behavior have greater odds of feeling unsafe at school (OR 1.39, 95% CI 1.24–1.57).

## 4. Discussion

### 4.1. Bullying victimization

The study on bullying victimization reveals several critical insights into the prevalence and determinants of bullying among different demographics and regions in China. Our study indicates that 19.4% of respondents reported experiencing bullying victimization, which is notably higher compared to a recent nationwide study in China that reported a bullying

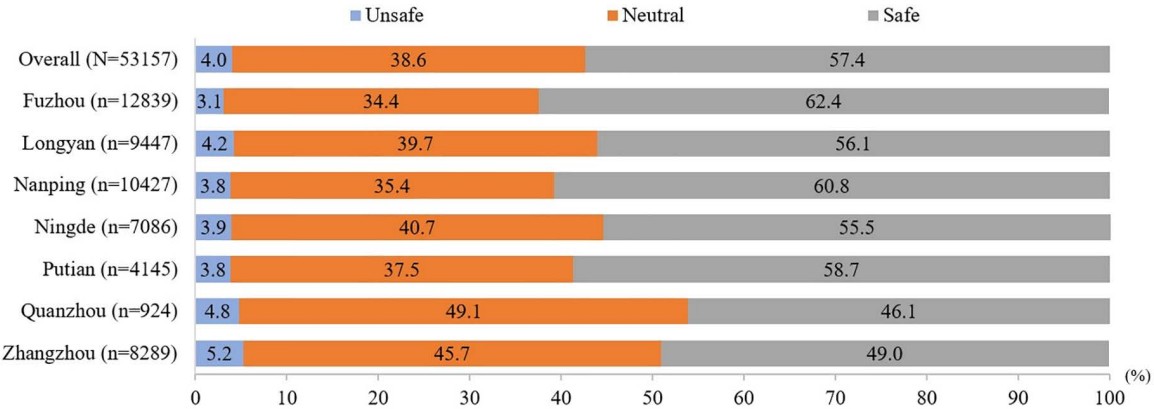

**Fig 2. Proportion of feeling safe at school by districts (N = 53157).**

victimization rate of 10.9% [4]. Geographically, the results show that the victimization rate of school bullying is higher in rural areas than in urban areas. Furthermore, the regional differences in bullying rates observed in this study may reflect local sociocultural factors, uneven distribution of educational resources, school safety management levels, or community support systems. Future research could further investigate the mechanisms of these regional factors.

A previously published study using the Global School-based Student Health Survey (GSHS) database found a positive association between social poverty indicators and bullying victimization at the country level, with higher levels of absolute social poverty linked to increased bullying [23]. In contrast, our study did not find a clear economic development trend but did find that higher parental education was associated with lower bullying victimization. This perhaps suggests that individual or micro-level disparities between poor and rich families might play a significant role at the district level. Another possible explanation for this heterogeneity is the use of subjective measures of bullying victimization in the current study. Differences in the measurement of bullying across studies could also contribute to the observed disparities. Additionally, it is also possible that factors beyond economic status, such as school environments [24] and community of social norms [25] which are not assessed in this study, may influence bullying rates.

The findings regarding parental educational level and bullying victimization are noteworthy. It was observed that participants whose parents had postgraduate degrees reported the lowest rates of bullying, suggesting the influence of socio-economic and individual-level factors on bullying victimization. Interestingly, while both paternal and maternal education showed significance in the univariate analyses, only maternal education remained a significant predictor in the multivariable model. This perhaps indicates the crucial role of maternal influence or involvement in children's social environments and coping mechanisms [26]. An important observation is the inverse correlation between age and bullying victimization. This implies that younger adolescents are at a higher risk of being bullied, possibly due to factors such as lower social skills, smaller physical stature, and lower tolerance for stress among younger cohorts [27]. Additionally, the study highlights the substantial impact of gender on bullying behavior, with boys exhibiting higher prevalence rates compared to girls, consistent with findings from various other studies [28,29]. Bullying victimization has consistently been linked to poor academic achievement in studies worldwide, including those conducted in China [6,30] and similar findings were observed in this study. Previous studies have reasoned that students who experience bullying often face academic difficulties, potentially due to factors such as increased stress, decreased self-esteem, disruptions in their learning environment, and diminished educational motivation caused by the bullying experience [30,31].

The current study revealed that a notable proportion of participants experienced depressive (19.1%), anxiety (22.0%), and stress (9.2%) symptoms. Additionally, psychological distress was identified in slightly over half of the participants

**Table 2. Factors associated with feeling safe at school (N = 53157).**

| | N (%) | Univariable analysis | | p-value | Multivariable analysis |
| | | Neutral/Safe (n = 51055) | Unsafe (n = 2102) | | Unsafe vs Neutral/Safe |
|---|---|---|---|---|---|
| | | | | | OR (95% CI) |
| Age (years) | | | | | |
| 12-14 | 21733 (40.9) | 21022 (96.7) | 711 (3.3) | p<0.001 | Reference |
| 15-16 | 20220 (38.0) | 19292 (95.4) | 928 (4.6) | | 1.44 (1.30-1.60)*** |
| 17-19 | 11204 (21.1) | 10741 (95.9) | 463 (4.1) | | 1.40 (1.24-1.58)*** |
| Gender | | | | | |
| Male | 27137 (51.1) | 25774 (95.0) | 1363 (5.0) | p<0.001 | 1.52 (1.38-1.67)*** |
| Female | 26020 (48.9) | 25281 (97.2) | 739 (2.8) | | Reference |
| Location | | | | | |
| Rural | 24223 (45.6) | 23146 (95.6) | 1077 (4.4) | p<0.001 | 1.02 (0.88-1.18) |
| Town | 20879 (39.3) | 20165 (96.6) | 714 (3.4) | | 0.97 (0.84-1.12) |
| City | 8055 (15.2) | 7744 (96.1) | 311 (3.9) | | Reference |
| Academic performance | | | | | |
| Top 15% (A level) | 8932 (16.8) | 8633 (96.6) | 300 (3.4) | p<0.001 | Reference |
| 15%−30% (B level) | 12664 (23.8) | 12294 (97.1) | 370 (2.9) | | 0.87 (0.75-1.02) |
| 30%−50% (C level) | 15779 (29.7) | 15245 (96.6) | 534 (3.4) | | 0.99 (0.86-1.15) |
| 50%−75% (D level) | 10938 (20.6) | 10491 (95.9) | 447 (4.1) | | 1.08 (0.93-1.26) |
| >75% (E level) | 4844 (9.1) | 4393 (90.7) | 451 (9.3) | | 2.31 (1.98-2.71)*** |
| Maternal highest education level | | | | | |
| Elementary School | 7466 (14.0) | 7101 (95.1) | 365 (4.9) | p<0.001 | Reference |
| Junior High School | 10500 (19.8) | 10066 (95.9) | 434 (4.1) | | 0.98 (0.80-1.19) |
| High School or Junior College | 19109 (35.9) | 18454 (96.6) | 655 (3.4) | | 0.90 (0.75-1.09) |
| Post-Secondary | 9251 (17.4) | 8905 (96.3) | 346 (3.7) | | 1.02 (0.83-1.26) |
| Undergraduate | 3285 (6.2) | 3171 (96.5) | 114 (3.5) | | 0.94 (0.72-1.24) |
| Postgraduates | 3546 (6.7) | 3358 (94.7) | 188 (5.3) | | 1.59 (1.21-2.08)** |
| Paternal highest education level | | | | | |
| Elementary School | 7389 (13.9) | 7027 (95.1) | 362 (4.9) | p<0.001 | 1.14 (0.89-1.49) |
| Junior High School | 7743 (14.6) | 7388 (95.4) | 355 (4.6) | | 1.20 (0.94-1.52) |
| High School or Junior College | 19778 (37.2) | 19085 (96.5) | 693 (3.5) | | 0.95 (0.76-1.19) |
| Post-Secondary | 10178 (19.1) | 9816 (96.4) | 362 (4.6) | | 0.91 (0.73-1.14) |
| Undergraduate | 3515 (6.6) | 3388 (96.4) | 127 (4.5 | | 0.96 (0.74-1.24) |
| Postgraduates | 4554 (8.6) | 4351 (96.5) | 203 (4.5) | | Reference |
| Family composition | | | | | |
| Intact Families | 47986 (90.3) | 46174 (96.2) | 1812 (3.8) | p<0.001 | Reference |
| Single-parent families | 1763 (3.3) | 1674 (95.0) | 89 (5.0) | | 1.17 (0.93-1.47) |
| Divorced families | 2556 (4.8) | 2419 (94.6) | 137 (5.4) | | 1.38 (1.15-1.67)** |
| Widowed families | 441 (0.8) | 407 (92.3) | 34 (7.7) | | 1.90 (1.31-2.76)** |
| Intergenerational families | 441 (0.8) | 381 (92.7) | 30 (7.3) | | 1.40 (0.94-2.09) |
| Experience with bullying | | | | | |
| Bullying victimization | | | | | |
| No | 42833 (80.6) | 41891 (97.8) | 942 (2.2) | p<0.001 | Reference |
| Yes | 10324 (19.4) | 9164 (88.8) | 1160 (11.2) | | 4.60 (4.14-5.11)*** |

*(Continued)*

**Table 2.** (Continued)

| | | Univariable analysis | | | Multivariable analysis |
|---|---|---|---|---|---|
| Bullying behavior | | | | | |
| No | 47885 (90.1) | 46407 (96.9) | 1478 (3.1) | p<0.001 | Reference |
| Yes | 5272 (9.9) | 4648 (88.2) | 624 (11.8) | | 1.39 (1.24-1.57)*** |

*p<0.05, **p<0.01, ***p<0.001.

Hosmer–Lemeshow test, chi-square: 12.714, p-value: 0.122; Nagelkerke $R^2$: 0.124.

(54.4%). When compared to a study conducted among high school students in Henan Province, where the bullying rate was 42.4%, the prevalence of depression among bullied students reached 30% Many studies, including review papers and research conducted in China, have demonstrated a link between poor mental health and bullying [4,8,32]. The findings from the multivariable analysis further underscore the robust association between various dimensions of psychological distress and bullying victimization. Specifically, all the three subscales of the DASS-Y reveal a gradient relationship where the severity of emotional state correlates with an increased likelihood of experiencing bullying. In addition, overall psychological distress as measured by the GHQ score was also significantly associated with bullying victimization. Individuals with higher GHQ scores were found to have higher odds of experiencing bullying compared to those with lower scores. The study highlights the substantial impact of bullying on mental health. It is important to note, however, that as a cross-sectional study, it cannot establish a causal relationship between bullying and depressive symptoms. This underscores the critical need for interventions targeting mental health to potentially mitigate the risk of bullying. Schools, parents, and mental health professionals should be attentive to the heightened vulnerability of individuals experiencing severe psychological distress and consider implementing supportive measures.

### 4.2. Bullying behavior

Our study indicates that 9.9% of respondents reported engaging in bullying behavior, which is relatively higher than the 1.38% reported in a previous study [4]. The distribution of bullying behavior by district is similar to that of bullying victimization. This study demonstrates that a significant majority of those who engage in bullying behavior have also experienced bullying themselves. Individuals who experience bullying victimization often exhibit bullying behavior themselves, a phenomenon known as the "victim-offender overlap" or "bully-victim overlap." Research indicates that those who are bullied are more likely to engage in aggressive behavior, perpetuating a cycle of violence and harm [33]. Therefore, it is imperative to implement interventions aimed at victims of bullying can significantly reduce bullying behavior.

Similar to findings on bullying victimization, bullying behavior was more prevalent among younger age groups, males, individuals in rural areas, those with lower academic performance, and those whose parents had lower levels of education. The mental health effects on those who bully others are not as thoroughly researched as the effects on victims. However, existing evidence suggests that engaging in bullying behavior is also linked to an elevated risk of various mental health issues [9]. Like victims, perpetrators may experience long-term negative impacts on their psychological well-being. This is supported by the current study, which found that the severity of psychological distress was correlated with an increased likelihood of bullying behavior. Overall psychological distress, as measured by the GHQ score, was significantly associated with bullying behavior, mirroring its relationship with bullying victimization.

### 4.3. Feeling safe at school

The study reveals that the majority of respondents feel safe at school, However, a considerable proportion of 38.6% remain neutral about their safety, indicating potential areas for improvement in creating a more reassuring atmosphere.

More importantly, the 3.9% of students who feel unsafe require immediate attention. A positive school climate that ensures both physical and emotional safety is essential for fostering a supportive learning environment where students feel secure, which results in enhancing their academic and personal development [34,35]. Similarly, the current study showed that not feeling safe at school was significantly associated with bullying victimization. However, it is interesting to note that our study also found that not feeling safe at school was significantly associated with bullying behavior. This could be a reaction of defense where students, feeling threatened, may resort to bullying others to protect themselves. Our findings imply that the perception of safety at school is a critical factor in understanding the dynamics of bullying, as it can influence both the likelihood of being a victim and the likelihood of engaging in bullying behavior. The findings suggest that a school climate that fosters feelings of safety and security can help reduce the incidence of bullying, both as a victim and as a perpetrator. Implementing targeted interventions such as anti-bullying programs, peer support networks, and increased security measures can help alleviate these concerns and create a safer environment for all students.

## 5. Implication for practice

The implications drawn from the study's findings on bullying victimization, bullying behavior, and feelings of safety at school underscore several critical points for practice and policy. Firstly, the significantly higher prevalence of bullying victimization among younger adolescents and those in rural areas highlights the need for targeted interventions early in schooling and specific geographic contexts. These interventions should focus not only on direct prevention of bullying incidents but also on fostering supportive environments that mitigate risk factors associated with bullying, such as socio-economic disparities and educational inequalities observed across different districts. Furthermore, the identified link between parental education and both victimization and perpetration suggest the potential for family-based interventions to play a pivotal role in reducing bullying behavior.

Secondly, the similarity observed between being bullied and engaging in bullying behavior emphasizes the importance of comprehensive anti-bullying strategies that address the broader psychological and social dynamics within schools. Implementing evidence-based programs that promote empathy, conflict resolution skills, and positive peer interactions can help break the cycle of bullying perpetration and victimization. Additionally, enhancing school safety measures and ensuring a positive school climate where students feel secure are crucial steps in fostering environments that discourage bullying behavior and support the well-being of all students. These efforts should also be complemented by mental health support services to address the emotional impact on both victims and perpetrators.

## 6. Limitation

While this study has several strengths such as a robust sample size, rigorous data collection methods, and comprehensive coverage of Fujian province, it also encountered several limitations. The potential bias in this sampling method could stem from the non-random selection of cities based on economic development levels, which may not accurately represent the entire province's diversity. While cities were randomly selected from each level, inherent differences within cities of the same level might exist, which could affect the representativeness of the sample. Despite the use of simple random sampling, where a secondary school and a high school were selected from each administrative area, respectively, the sampling within pre-selected administrative areas and schools might overlook unique local variations. Additionally, the third stage of cluster sampling, while aiming to ensure adequate representation of different grades, may still result in underrepresentation or overrepresentation of certain classes, depending on the distribution of students within schools. Furthermore, selecting a fixed percentage of classes within schools may lead to unequal representation of smaller or larger schools. These biases could affect the generalizability of the study's findings to all adolescents in Fujian Province. It is also important to note that, the data relied on self-reported data, which may be subject to recall bias. In addition, this is a cross-sectional study, and the causal relationship cannot be determined. Additionally, the administration of questionnaires by classroom teachers may have influenced the honesty of students' responses,

particularly on sensitive topics such as bullying. Future studies could consider using more neutral third-party administrators or anonymous online data collection to reduce social desirability bias. It should also be noted that we did not adjust for clustering effects at the school/class level in our regression analysis, which may have led to an underestimation of the standard error. Last limitation of the study is its cross-sectional design, which may not allow for establishing a clear causal relationship between the variables.

## 7. Conclusion

This study on bullying victimization provides valuable insights into its prevalence and underlying factors across different demographics and regions in the Fujian Province, China. The findings highlight a concerning rate of bullying victimization. Geographic disparities reveal varying rates across districts, underscoring the complex nature of bullying influenced by local contexts rather than straightforward economic development indicators. Parental education emerges as a significant factor, suggesting potential avenues for intervention targeting socio-economic disparities. The study also underscores the vulnerability of younger adolescents, the impact of gender, and the relationship between bullying victimization and psychological distress. Concurrently, the study identified a concerning rate of bullying behavior, emphasizing the cyclical nature where victims may also become perpetrators. This victim-perpetrator cycle underscores the urgent need for integrated anti-bullying strategies that address both victimization and perpetration dynamics. More importantly, both bullying victimization and behaviors are strongly associated to poor psychological health. Additionally, the study underscores the critical link between feelings of safety at school and bullying involvement, demonstrating that students who feel unsafe are more likely to experience or engage in bullying behavior. The findings also underscore the importance of interventions that target socio-economic disparities across diverse geographic regions.

## Supporting information

**S1 Checklist. Questionnaire on inclusivity in global research.** The completed PLOS questionnaire regarding ethical, cultural, and scientific considerations for this research.
(DOCX)

## Acknowledgments

The authors would like to thank the schools, teachers, students and parents who participated in this study for their help and support and the students for completing the survey in a complete, honest and detailed manner.

## Author contributions

**Conceptualization:** Jian Jiang.

**Data curation:** Jian Jiang.

**Formal analysis:** Zhijie Luo, Yuhang Chen, Haridah Alias.

**Funding acquisition:** Zhijian Hu.

**Methodology:** Jian Jiang, Zhijie Luo, Yuhang Chen, Haridah Alias.

**Project administration:** Li Ping Wong, Yulan Lin, Zhijian Hu.

**Supervision:** Zhijian Hu.

**Writing – original draft:** Jian Jiang.

**Writing – review & editing:** Li Ping Wong, Yulan Lin, Zhijian Hu.

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
