## [Decision Letter · Decision Letter 0]

3 Feb 2026

Dear Dr. Jiang,

Thank you for submitting your manuscript to PLOS ONE. After careful consideration, we feel that it has merit but does not fully meet PLOS ONE’s publication criteria as it currently stands. Therefore, we invite you to submit a revised version of the manuscript that addresses the points raised during the review process.

A letter that responds to each point raised by the academic editor and reviewer(s). You should upload this letter as a separate file labeled ‘Response to Reviewers’.A marked-up copy of your manuscript that highlights changes made to the original version. You should upload this as a separate file labeled ‘Revised Manuscript with Track Changes’.An unmarked version of your revised paper without tracked changes. You should upload this as a separate file labeled ‘Manuscript’.

We look forward to receiving your revised manuscript.

Kind regards,

Alejandro Botero Carvajal, Ph.D

Academic Editor

PLOS One

Journal Requirements:

1. Please ensure that your manuscript meets PLOS ONE’s style requirements, including those for file naming. The PLOS ONE style templates can be found at

This work was supported by the Scientific and Technological Innovation Joint Capital Projects of Fujian Province (No.2020Y9018); the Natural Science Foundation of Fujian Province (No. 2021J01726, No. 2021J01733); the Central government-led local science and technology development special project (No. 2020L3009, No. 2019L3006), China.

4. In this instance it seems there may be acceptable restrictions in place that prevent the public sharing of your minimal data. However, in line with our goal of ensuring long-term data availability to all interested researchers, PLOS’ Data Policy states that authors cannot be the sole named individuals responsible for ensuring data access (http://journals.plos.org/plosone/s/data-availability#loc-acceptable-data-sharing-methods).

Reviewers’ comments:

Reviewer's Responses to Questions

**Comments to the Author**

1. Is the manuscript technically sound, and do the data support the conclusions?

Reviewer #1: Yes

Reviewer #2: Yes

2. Has the statistical analysis been performed appropriately and rigorously?

Reviewer #1: Yes

Reviewer #2: Yes

3. Have the authors made all data underlying the findings in their manuscript fully available?

Reviewer #1: Yes

Reviewer #2: Yes

4. Is the manuscript presented in an intelligible fashion and written in standard English?

Reviewer #1: No

Reviewer #2: Yes

Reviewer #1: 1- The reason why a city has more bullying rate compared to other cities? Any specific reason?

2- A limitation of the study is that the questionnaire was conducted by teachers: some students may not feel comfortable saying the truth to their teacher.

3- the discussion is very long and confusing, make it concise and to the point.

Reviewer #2: Results:

Results were reported in detail. However there are some wording mistakes:

1. Under 3.2 Bullying Behavior, last line in paragraph 2:

“Based on academic achievement, the odds of experiencing bullying were 1.22 times higher for participants with D-level performance (OR 1.22, 95% CI 1.10 1.34) and 1.40 times higher for those with E-level performance (OR 1.40, 95% CI 1.25-1.57) compared to those in the top 15% (A level).”

2. Under 3.2 Bullying Behavior, last line in paragraph 4:

“Individuals with a GHQ score of 3-12 had 1.71 times higher odds of bullying victimization compared to those with a score of 0-2 (OR 1.71, 95% CI 1.60-1.83).”

Comment: Author used “experiencing bullying” and “bullying victimization” as the variable but the result is about bullying behavior not bullying victimization.

3. Added suggestion: In 3.3 Feeling Safe in School, author describes table 2 showing the association between bullying victim/behavior and feeling safe. However, the table also showed many other important factors. Suggest changing the sentence to: “The associated factors with feeling safe in school were shown in Table 2”

Discussion:

Well written.

.

Reviewer #1: No

Reviewer #2: No

---

## [Author Response · Author response to Decision Letter 1]

9 Feb 2026

Response to Reviewers

Academic Editor

Requirements 1:

Please ensure that your manuscript meets PLOS ONE's style requirements, including those for file naming. The PLOS ONE style templates can be found at https://journals.plos.org/plosone/s/file?id=wjVg/PLOSOne_formatting_sample_main_body.pdf and https://journals.plos.org/plosone/s/file?id=ba62/PLOSOne_formatting_sample_title_authors_affiliations.pdf

Response:

Thank you for reminding us to ensure that our manuscript meets PLOS ONE’s style requirements. We have carefully reviewed and followed the journal’s style templates.

Requirements 2:

Please include a complete copy of PLOS’ questionnaire on inclusivity in global research in your revised manuscript. Our policy for research in this area aims to improve transparency in the reporting of research performed outside of researchers’ own country or community. The policy applies to researchers who have travelled to a different country to conduct research, research with Indigenous populations or their lands, and research on cultural artefacts. The questionnaire can also be requested at the journal’s discretion for any other submissions, even if these conditions are not met. Please find more information on the policy and a link to download a blank copy of the questionnaire

here: https://journals.plos.org/plosone/s/best-practices-in-research-reporting. Please upload a completed version of your questionnaire as Supporting Information when you resubmit your manuscript.

Response:

We have downloaded and completed the full version of the PLOS’ questionnaire on inclusivity in global research inclusion, and uploaded it with the manuscript.

Requirements 3:

Thank you for stating the following financial disclosure:

This work was supported by the Scientific and Technological Innovation Joint Capital Projects of Fujian Province (No.2020Y9018); the Natural Science Foundation of Fujian Province (No. 2021J01726, No. 2021J01733); the Central government-led local science and technology development special project (No. 2020L3009, No. 2019L3006), China.

Response:

Thank you very much for your suggestion. We reiterated in our cover letter that "The funders had no role in study design, data collection and analysis, decision to publish, or preparation of the manuscript."

Requirements 4:

In this instance it seems there may be acceptable restrictions in place that prevent the public sharing of your minimal data. However, in line with our goal of ensuring long-term data availability to all interested researchers, PLOS’ Data Policy states that authors cannot be the sole named individuals responsible for ensuring data access (http://journals.plos.org/plosone/s/data-availability#loc-acceptable-data-sharing-methods).

Response:

Indeed, thank you very much for your suggestion. To ensure the accessibility of our research data, we have added information on data availability to the "Availability of data and materials" section of the manuscript: "If needed for the research, data may be obtained from all corresponding authors or the School of Public Health of Fujian Medical University."

Requirements 5:

Please include captions for your Supporting Information files at the end of your manuscript, and update any in-text citations to match accordingly. Please see our Supporting Information guidelines for more information: http://journals.plos.org/plosone/s/supporting-information.

Response:

Thank you very much, we have no additional information.

Requirements 6:

Response:

Thank you very much. The reviewers did not suggest that we cite any previously published works.

Requirements 7:

Response:

Thank you for your important comment regarding the completeness and correctness of the reference list. We have conducted a systematic review and correction of all references in the manuscript in strict accordance with your guidance.

Reviewer 1

Comments 1:

The reason why a city has more bullying rate compared to other cities? Any specific reason?

Response:

Thank you for raising this important point. Our study indeed observed variations in bullying rates across cities. These differences may be attributable to regional socioeconomic development, educational resource allocation, school culture, community norms, and family support systems. Although our study did not directly measure these factors, we have mentioned the potential influence of “school environments” and “community social norms” in the Discussion (Section 4.1). We will elaborate on this further and suggest future research to explore the underlying mechanisms of regional disparities.

Comments 2:

A limitation of the study is that the questionnaire was conducted by teachers: some students may not feel comfortable saying the truth to their teacher.

Response:

We fully agree with this observation. Although we implemented several quality control measures (e.g., trap questions, anonymous responses), the presence of teachers may still influence students' willingness to answer truthfully. We will explicitly acknowledge this limitation in the “Limitations” section (Section 6).

Comments 3:

the discussion is very long and confusing, make it concise and to the point.

Response:

Thank you for this constructive suggestion. We will streamline the Discussion section by removing repetitive content, merging similar paragraphs, and strengthening the logical flow. See the "Discussion" section

Reviewer 2

Comments 1:

Results were reported in detail. However there are some wording mistakes:

1. Under 3.2 Bullying Behavior, last line in paragraph 2:

“Based on academic achievement, the odds of experiencing bullying were 1.22 times higher for participants with D-level performance (OR 1.22, 95% CI 1.10 1.34) and 1.40 times higher for those with E-level performance (OR 1.40, 95% CI 1.25-1.57) compared to those in the top 15% (A level).”

2. Under 3.2 Bullying Behavior, last line in paragraph 4:

“Individuals with a GHQ score of 3-12 had 1.71 times higher odds of bullying victimization compared to those with a score of 0-2 (OR 1.71, 95% CI 1.60-1.83).”

Comment: Author used “experiencing bullying” and “bullying victimization” as the variable but the result is about bullying behavior not bullying victimization.

Response:

Thank you for catching this terminology inconsistency. We have corrected the wording.

Comments 2:

Added suggestion: In 3.3 Feeling Safe in School, author describes table 2 showing the association between bullying victim/behavior and feeling safe. However, the table also showed many other important factors. Suggest changing the sentence to: “The associated factors with feeling safe in school were shown in Table 2”.

Response:

Thank you for this thoughtful suggestion. We agree that the original sentence did not fully reflect the multiple factors presented in Table 2, which includes age, gender, academic performance, parental education, and others. We will revise the sentence accordingly to provide a more accurate summary of the table.

---

## [Decision Letter · Decision Letter 1]

18 Mar 2026

Dear Dr. Jiang,

Thank you for submitting your manuscript to PLOS ONE. After careful consideration, we feel that it has merit but does not fully meet PLOS ONE’s publication criteria as it currently stands. Therefore, we invite you to submit a revised version of the manuscript that addresses the points raised during the review process.

Overall, the manuscript demonstrates:

A well-defined and justified research question supported by relevant literature.

A clear and appropriate observational study design with comprehensive methodological description.

Statistical analyses that are suitable for the research aims and correctly interpreted.

Conclusions that are consistent with the data and appropriately cautious regarding the cross-sectional nature of the study.

Transparent reporting of ethical approval, funding, and data availability.

Required Revisions (Minor)

Clarify analytic approach regarding cluster sampling

Although the sampling strategy employed a multistage cluster design, the regression analyses do not appear to adjust for clustering at the school/class level.

Please explicitly acknowledge this in the Limitations section, noting the potential for underestimated standard errors.

Tidy remaining editorial elements

A few minor textual issues remain (e.g., repeated lines in the Data Availability statement and minor typographical inconsistencies).

Please revise for clarity and consistency.

Optional but recommended:

Briefly reinforce in the Limitations that teacher-administered questionnaires may increase social desirability bias in sensitive topics (bullying, mental health).

We look forward to receiving your revised manuscript.

Kind regards,

Alejandro Botero Carvajal, Ph.D

Academic Editor

PLOS One

Journal Requirements:

Reviewers' comments:

Reviewer's Responses to Questions

**Comments to the Author**

Reviewer #2: All comments have been addressed

2. Is the manuscript technically sound, and do the data support the conclusions?

Reviewer #2: Yes

3. Has the statistical analysis been performed appropriately and rigorously?

Reviewer #2: I Don't Know

4. Have the authors made all data underlying the findings in their manuscript fully available?

Reviewer #2: Yes

5. Is the manuscript presented in an intelligible fashion and written in standard English?

Reviewer #2: Yes

Reviewer #2: (No Response)

.

Reviewer #2: No

---

## [Author Response · Author response to Decision Letter 2]

19 Mar 2026

Response to Reviewers

Required Revisions (Minor)

Requirements 1:

Clarify analytic approach regarding cluster sampling

Although the sampling strategy employed a multistage cluster design, the regression analyses do not appear to adjust for clustering at the school/class level.

Please explicitly acknowledge this in the Limitations section, noting the potential for underestimated standard errors.

Response:

We thank the reviewer for highlighting this important methodological concern. We agree that failing to adjust for clustering effects in a multistage cluster sampling design may lead to underestimated standard errors and affect the precision of statistical inferences. In response, we have added the following statement to the Limitation section (Section 6):

“It should also be noted that we did not adjust for clustering effects at the school/class level in our regression analysis, which may have led to an underestimation of the standard error.”

Requirements 2:

Tidy remaining editorial elements

A few minor textual issues remain (e.g., repeated lines in the Data Availability statement and minor typographical inconsistencies).

Please revise for clarity and consistency.

Response:

We thank the reviewer for their careful reading. We have thoroughly reviewed the manuscript and made the following corrections:

Data Availability Statement: Duplicate lines have been removed to ensure clarity and conciseness.

Typographical inconsistencies: We have corrected punctuation, spacing, capitalization, and other minor formatting issues throughout the text.

Terminology consistency: We have ensured that the terms “bullying victimization” and “bullying behavior” are used consistently across the manuscript (as previously suggested by reviewers).

Requirements 3:

Optional but recommended:

Briefly reinforce in the Limitations that teacher-administered questionnaires may increase social desirability bias in sensitive topics (bullying, mental health).

Response:

We thank the reviewer for this valuable suggestion. We have added the following statement to the Limitations section (Section 6)：

“Additionally, the administration of questionnaires by classroom teachers may have influenced the honesty of students’ responses, particularly on sensitive topics such as bullying.”

---

## [Editor Report · Decision Letter 2]

22 Mar 2026

Dear Dr. Jiang,

Thank you for submitting your manuscript to PLOS ONE. After careful consideration, we feel that it has merit but does not fully meet PLOS ONE’s publication criteria as it currently stands. Therefore, we invite you to submit a revised version of the manuscript that addresses the points raised during the review process.

Before final acceptance, please address the following minor points to ensure internal consistency and compliance with journal standards:

Sample size consistency

Please reconcile the discrepancy in the reported sample size between the Abstract and the Results/Tables. If exclusions were applied after initial screening, clearly state this and ensure the same final N is used consistently throughout the manuscript.

Consistency between text and tables

Carefully review the Results section to ensure that all odds ratios, confidence intervals, and p-values reported in the text exactly match those presented in the corresponding tables. Any discrepancies should be corrected.

Data Availability statement

Please revise the Data Availability statement to align fully with journal policy by specifying a clear institutional access pathway for data requests. Avoid phrasing such as “available from the authors upon request” and instead indicate the responsible institutional body and conditions for access.

Minor editorial and formatting issues

Perform a final careful language and formatting check to correct minor typographical errors, spacing inconsistencies, and ensure uniform presentation of abbreviations, percentages, and decimal places across the manuscript and tables.

We look forward to receiving your revised manuscript.

Kind regards,

Alejandro Botero Carvajal, Ph.D

Academic Editor

PLOS One
---

## [Author Response · Author response to Decision Letter 3]

23 Mar 2026

Requirements 1:

Sample size consistency

Please reconcile the discrepancy in the reported sample size between the Abstract and the Results/Tables. If exclusions were applied after initial screening, clearly state this and ensure the same final N is used consistently throughout the manuscript.

Response:

A total of 54,368 questionnaires were initially collected. During the data cleaning stage, we applied predefined quality control criteria (including trap question screening and removal of straight-lining responses) to exclude invalid questionnaires. Consequently, the final analytic sample comprised 53,157 valid responses, which is the sample size used in all tables (Table 1 and Table 2).

To ensure consistency, we have made the following revisions:

1. Abstract: Changed “Analysis of 54,368 responses” to “Analysis of 53,157 valid responses” to reflect the final analytic sample.

2. Results, first paragraph: Changed “A total of 54,368 complete responses were included” to “A total of 54,368 questionnaires were collected. After data cleaning, 53,157 valid responses were included in the final analysis.”

3. Table titles: Table 1 and Table 2 now clearly indicate N=53,157, consistent with the text.

4. Other descriptions: All mentions of sample size throughout the manuscript have been reviewed and revised to consistently use the final analytic sample of 53,157.

Requirements 2:

Consistency between text and tables

Carefully review the Results section to ensure that all odds ratios, confidence intervals, and p-values reported in the text exactly match those presented in the corresponding tables. Any discrepancies should be corrected.

Response:

Thank you for your careful review regarding the consistency between the text and the tables. We acknowledge that there were discrepancies in the reporting of some data when citing from the tables into the text, which occurred due to clerical errors. We have now thoroughly verified all data points and made the necessary revisions to ensure that all odds ratios, confidence intervals, and p values reported in the text are fully consistent with those presented in the corresponding tables. All modifications have been marked using track changes in the revised manuscript for your reference. We appreciate your attention to detail and apologize for any confusion caused.

Requirements 3:

Data Availability statement

Please revise the Data Availability statement to align fully with journal policy by specifying a clear institutional access pathway for data requests. Avoid phrasing such as “available from the authors upon request” and instead indicate the responsible institutional body and conditions for access.

Response:

Thank you for the guidance on revising the Data Availability statement to align with PLOS ONE policy. We have modified the statement as follows to ensure it specifies a clear institutional access pathway:

“Due to ethical and confidentiality concerns regarding adolescent participants, the research data are not publicly available. Data requests may be submitted to the Ethics Committee of Fuzhou University Affiliated Provincial Hospital (Email: fjslkyk@163.com), which will review requests in accordance with institutional ethical and privacy policies. Access will be granted to qualified researchers for purposes aligned with the study’s ethical approval.”

Requirements 4:

Minor editorial and formatting issues

Perform a final careful language and formatting check to correct minor typographical errors, spacing inconsistencies, and ensure uniform presentation of abbreviations, percentages, and decimal places across the manuscript and tables.

Response:

Thank you for your careful review of the editorial and formatting details. We have conducted a thorough language and formatting check throughout the manuscript. All revisions have been marked using track changes in the revised manuscript for your reference. We believe that the language and formatting of the manuscript now fully meet the journal’s requirements. Thank you again for your meticulous review.

---

## [Editor Report · Decision Letter 3]

25 Mar 2026

School bullying and its association with psychological wellbeing: findings of the Fujian Adolescent Mental Wellness Study (FAMWeS)

PONE-D-24-55844R3

Dear Dr. Jiang,

We’re pleased to inform you that your manuscript has been judged scientifically suitable for publication and will be formally accepted for publication once it meets all outstanding technical requirements.

Kind regards,

Alejandro Botero Carvajal, Ph.D

Academic Editor

PLOS One
---

## [Editor Report · Acceptance letter]

PONE-D-24-55844R3

PLOS One

Dear Dr. Jiang,

I'm pleased to inform you that your manuscript has been deemed suitable for publication in PLOS One. Congratulations! Your manuscript is now being handed over to our production team.

Kind regards,

on behalf of

Dr. Alejandro Botero Carvajal

Academic Editor

PLOS One